# Complex Geophysical Investigations under Extreme P,T–Conditions at Zentralinstitut für Physik der Erde (ZIPE) (1970–1990)

**Hans J. Mueller [1],\* and Heiner Vollstädt [2]**

[1]  Karlsruhe Institute of Technology, Institute of Applied Geosciences, Adenauerring 20b, building 50.40, D-76131 Karlsruhe, Germany

[2]  Vollstädt–Diamant GmbH, Kiefernweg 7, D-14554 Seddiner See, Germany; info@vollstaedt.com

\*  Correspondence: hans-joachim.mueller@rcftp.de; Tel.: +49-721-608-45221

**Abstract:** The development of the geophysical high pressure research in the former German Democratic Republic (GDR) is described here. The GDR was a German state established in 1949 at the territory of the Soviet occupation zone. The different experimental investigations under extreme pressure and temperature conditions and their industrial applications, including the pilot manufacture of synthetic diamonds are explained. A review of the research topics pursued including experiments on lunar material and Earth core/mantle material is described.

**Keywords:** high pressure; physical properties; partial melting; lunar samples; diamond synthesis

---

## 1. Introduction

High pressures and high temperatures are two of the most important physical parameters in characterizing the physics of materials. In chemistry research and production lines under higher pressures and temperatures are also important; but in this field the volumes are bigger and pressures over 1 GPa are rarely applied.

In the mid-1970s when the author began working at the Zentralinstitut für Physik der Erde (ZIPE), scientists could only read international journals at the institute's library. Based on these data we could fill out order cards for special printed copies. Any personal contact with the authors was not allowed. Some of these special printed copies became personal guides to the scientific. For us in the discipline, perhaps the most valuable were undoubtedly the papers by Birch (1960/1961), Christensen (1965), Manghnani et al. (1974) and Kern (1978) [1–5]. Everything that was described in this literature could also be possible at Potsdam–pressure, geologically targeted samples, ultrasonic measurements, material interpretation of seismic data. We have to realize the world was divided into two different scientific regions at that time. That means any international published result could be possible by inaccessible materials and equipment. Consequently this "translation" to the "eastern" market was the first step of commencing scientific work. The publication of our own research results only happened at meetings and in the journals of countries of the Council for Mutual Economic Assistance (COMECON—an "eastern" analogue of the European Economic Community– EEC). During the 1970s, the conditions and methods of ZIPE experiments became comparable to those in the western countries and achieved-especially with the new "French press" (see later) – even higher conditions. The ZIPE department V had close relations with geoscientific institutions inside COMECON mainly at Kiev, Moscow, Novosibirsk, Apatity and Prague, including mutual visits. Later in the mid-1980s, the publication in "western" scientific journals was allowed and became customary. Consequently, the years from 1989—the year of the peaceful revolution—to 1992—the year of establishing the

GeoForschungsZentrum Potsdam—were, for the ZIPE scientific staff, the first time to be involved in the worldwide science community.

It was one of the peculiarities of scientific research in the German Democratic Republic (GDR) that the main part of high pressure research was conducted in the field of geosciences. From the mid-1960s to the early 1970s, the Institute for Geodynamik in Jena and the Geomagnetic Institute in Potsdam (integrated in the later Central Institute for Physics of the Earth) and the Karl Marx University in Leipzig, initiated the first experimental high pressure research. The background and basis for these research activities were polymorphic [6–9].

In Potsdam, at the Geomagnetic Institute, a rock magnetic department was established under the aegis of Frölich. The original theoretical and experimental investigations on the behaviour of the Earth's magnetic field and its geological development over time were very soon followed by the first rock-physical investigations under higher pressure and temperature conditions. The influence of Stiller and Vollstädt was particularly noticeable in this development; the latter was able to provide the mineralogical component to the mainly geophysically oriented group (see Table 1).

**Table 1.** The status of high pressure research at different institutions of the German Democratic Republic (GDR) in 1967.

| Institution | VEB Geophysik Leipzig | department for geophysical survey and geology of Karl-Marx-University Leipzig | Central Institute for Physics of the Earth, Potsdam | Central Institute for Physics of the Earth, Potsdam Physics section, Dep. of Crystallography of Humboldt-University Berlin |
|---|---|---|---|---|
| equipment | 200 t press | 100 t press | 200 t press, 50 t press, 2.5 t press | high pressure device, pressure chamber up to 40–60 kbar, uniaxial with small internal space |
| | ultrasonic equipment 0.3–1 MHz | ultrasonic equipment, high pressure hamber up to 8 kbar in test | ultrasonic equipment 1–4 MHz, high resistance measurement devices, strain gauges measurement devices, pressure chamber up to 2 kbar in test | polarisation microscopy, X-ray and spectroscopic devices |
| recent research | comparison of sound velocities of in-situ measurements with laboratory tests under uniaxial pressure | hysteresis and anisotropy of p-wave velocities under uniaxial pressure | deformation and electrical resistivity under uniaxial pressure | |
| research projects | pressure chamber up to 2 kbar | | pressure chamber up to 10 kbar with heater, pressure chamber up to 30 kbar with heater, pressure chamber up to 100 kbar with heater | further development above chamber (heating) |

In Jena, a strong theoretical group was established in the Institute for Geodynamics under the aegis of Uhlmann, which took particular interest in the condition of the deeper interior of the Earth and the material behaviour under extreme pressure and temperature conditions [10–13]. There too, under the influence of Stiller and against the background of theoretical investigations, a high-pressure group with a geophysical orientation similar to that in Potsdam was formed, i.e., mainly at conditions of the lower crust and upper mantle. However, reaching higher pressures was not given up. Later at Potsdam a new high pressure laboratory was established for performing experiments with the newly developed split sphere high pressure chamber (see later).

Organizational changes within the field of geosciences of the Academy of Sciences of the GDR led to the fusion of these regionally separated research activities and to the later concentration of the experimental activities at Potsdam (see Figures 1–4).

At the end of 1970 the first successful synthesis of industrial diamonds was made at ZIPE, for the time being without economic consequences. In 1981, the industrial pilot production was started in cooperation with district industry. By 1989, all hard material research and pilot production in Oranienburg were discontinued.

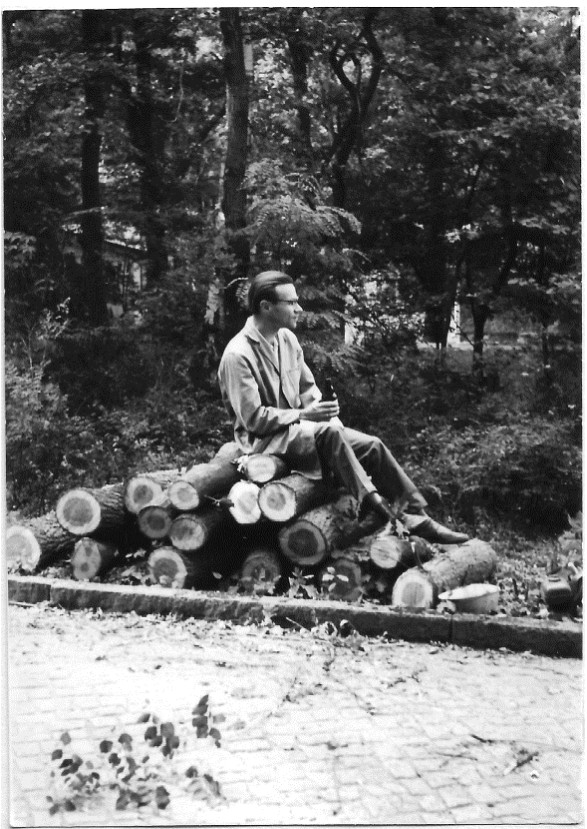

**Figure 1.** "Wood cutters" rest—von Faber.

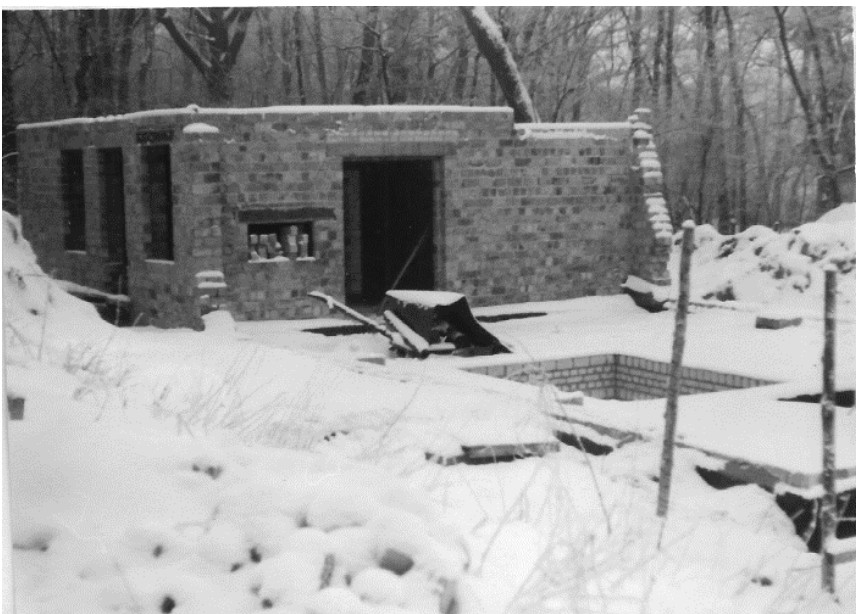

**Figure 2.** Construction site of the first high pressure hall with a pit for the triaxial press (6 January 1971).

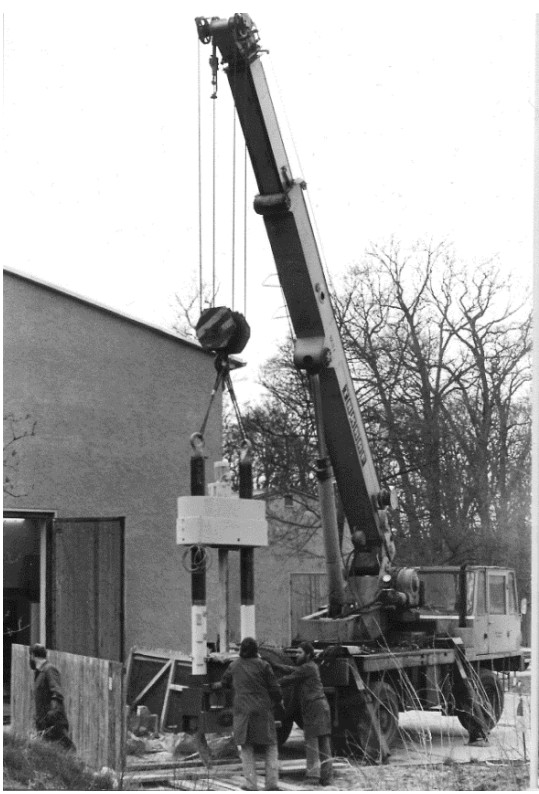

**Figure 3.** Assembling the triaxial press.

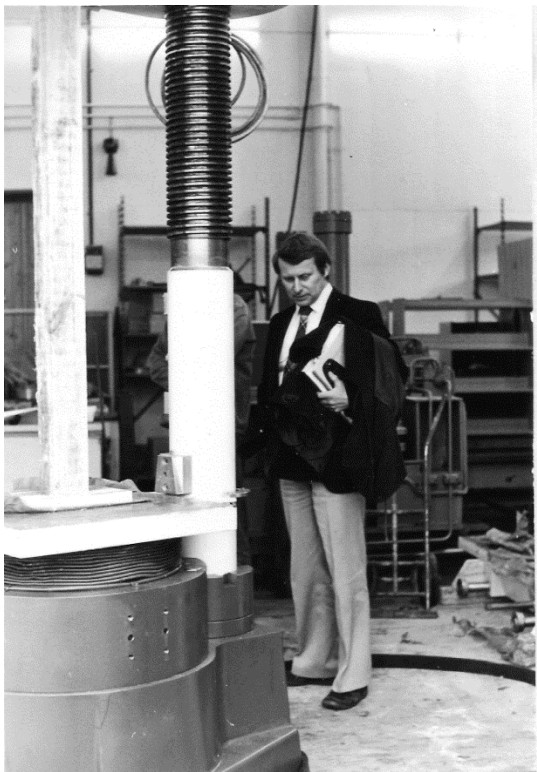

**Figure 4.** Prof. Vollstädt and his biggest press at that time.

## 2. Methods and Equipment of Measurement

### 2.1. Petrophysical Properties of Rocks and Minerals under Extreme P,T–Conditions

In autumn 1973, department V of ZIPE consisted of two buildings—a basement two-storey office part with lecture hall and the high-pressure laboratory, consisting of a hall, two offices and a large preparation room. In the hall there were two hydraulic presses—500 t and 600 t maximum load. For the latter, there was a removable carriage with a lateral punch ring allowing forces up to a maximum of 400 t. When combined this resulted in a triaxial system; i.e., a compression system in which a cube-shaped sample could be compressed in three directions. The world-famous laboratory of Prof. Kern in Kiel had the same principle. A piston heater was also developed in order to be able to experimentally simulate the temperature, which increased with depth, during the high-pressure experiments. After moving out the triaxial insert, a 600 t press with a yoke adjustable via two spindles was available [14–17] (see Figures 5 and 6).

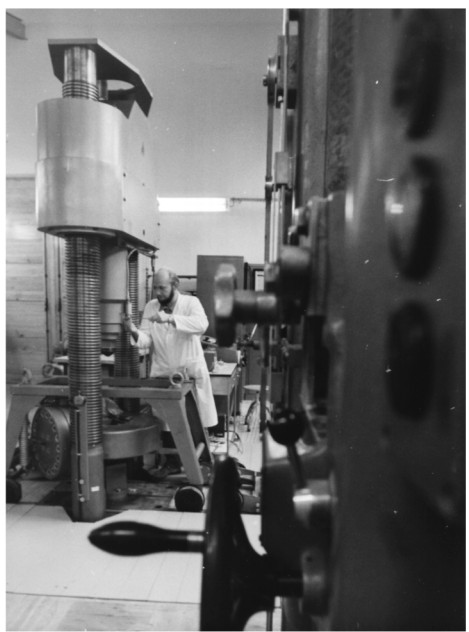

**Figure 5.** 600 t press with triaxial device.

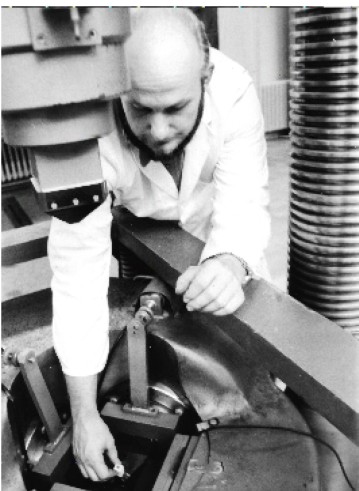

**Figure 6.** Sample insertion at the triaxial press.

The 500 t press had a manually adjustable yoke. Two classical oil pressure chambers up to 0.5 GPa maximum pressure, inner diameter 50 mm, as well as two piston-cylinder devices with different diameters (10 and 17 mm) of the ring system up to 1.75 GPa were available. Lead sheeting was used as a quasi-hydrostatic pressure transfer medium in which the rock samples were embedded. Ceramic ultrasonic transducers for compressional waves were installed in the upper and lower part of the column guide frames. For the oil pressure chambers there were initially no fixed measuring set-ups available. The task of the first author's (Mueller) final thesis was to develop an attenuation measurement for P-waves. The solution was a spring-loaded transmission method. Since the elastic coupling to the sample is critical in this case and the oil infiltration of samples should be avoided, all rock samples were covered by an air-drying lacquer. The ultrasonic transducers were separated from the sample only by a thin copper foil. The damping values were determined relative to a metallic specimen of known attenuation or by measuring with two samples of different lengths. In the course of the expansion of the high-pressure methods for measuring the elastic properties, a statistical measuring principle was established; i.e., first, the elastic wave velocities of all samples of a rock up to 0.5 GPa and room temperature were measured. On this basis, representative samples for the high-pressure, high-temperature experiments were then conducted, which were then finally incorporated into interpretation of deep seismic data. For this purpose, a special measuring set-up with two pairs of barium-titanate-zirconate ultrasonic transducers for elastic compression and shear waves was developed (see Figures 7 and 8). Measuring electronics, two measuring instruments for electric cables with preamplifiers, were utilized. In compact form, the instruments contained all the components (pulse generator, delay unit, oscilloscope) which are required for an ultrasonic measurement. Later a special ultrasonic measurement device from the company Krompholz was obtained [18–23].

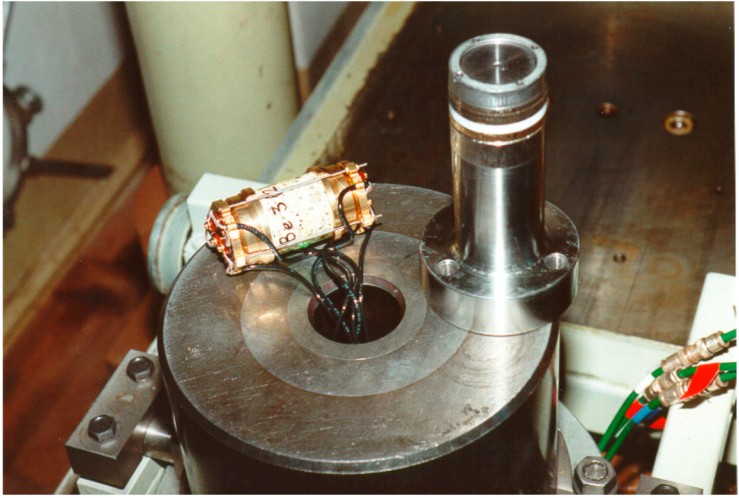

**Figure 7.** Internal set-up of the 0.5 GPa oil pressure chamber.

These complex physical laboratory investigations also included measurements of the electrical conductivity and thermal properties of rocks under ambient and extreme pressure and temperature conditions. The results showed that with increasing pressure e.g., the electrical resistance increases, whereby the increase is more significant for sediments than for magmatic rocks.

Other parameters, such as magnetic quantities (magnetic anisotropy, direction of magnetization, Curie temperature), were also the subject of laboratory investigations and were used to clarify the magnetic rock behaviour down to the pressures of the deeper Earth crust (about 1 GPa and 750 °C).

The existing high-pressure measurements of elastic properties were essentially limited to crustal pressures; i.e., hydrostatic 0.5 GPa and quasi-hydrostatic 0.6 GPa. Thus, if the conversion factor of pressure in GPa to depth in km was 30, this corresponded to depths less than 20 km. It was then

decided to purchase an oil pressure chamber up to a maximum pressure of 1.2 GPa. For this purpose, a two-layer shrunk jacketed vessel was built. Following the example of the existing 0.5 GPa chamber, the pressure chamber stood on the fixed lower punch with the electrical feedthroughs. The long upper punch was used for compression (Figures 7 and 8). The inner structure, the actual measuring cell, was connected to the lower punch with cables. Since there were problems with the compression path in practical test operation, the measuring assembly was connected to the lower punch with a plug. For this purpose, the measuring set-up was fed from above with a specially manufactured key and connected to the lower punch. However, there were problems with the pressure-transmitting medium. The hydraulic oil, which had been used up to 0.5 GPa, could no longer be used. From about 0.7 to 0.8 GPa its viscosity increased so much that cables were torn. Therefore, a mixture of ethanol and methanol was used as a new pressure transmitting medium. As expected, the viscosity behaviour was excellent, but the lubricating effects on the metal seals were not. Therefore, up to 10% hydraulic oil was added as lubricant. The measuring set-up was designed for measuring the elastic wave velocities of longitudinal and shear waves up to 750 °C [24–36].

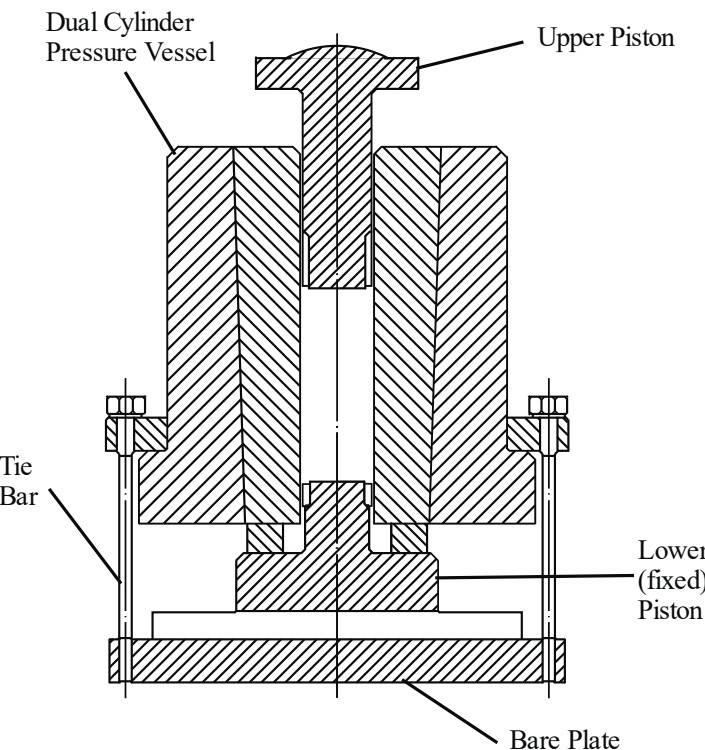

**Figure 8.** Sketch of the 1.2 GPa oil pressure chamber.

In 1977, there was an opportunity for scientific institutions to order major purchases on the world market, if this could remedy a massive shortage in the sense of scientific progress. Department V applied for a high-performance gas pressure chamber and was awarded the contract. Harwood, USA and Basset Bretagne Loire, France were shortlisted. Harwood offered a gas pressure chamber operated with argon up to 1.4 GPa. From Basset came a bid for a helium chamber up to 2.5 GPa (Figures 9 and 10). Therefore, the French offer was chosen. According to reports the selling price was 1.5 Mio Valuta Mark. In autumn 1977 a very large box arrived at Potsdam and was placed on the ramp of the high pressure laboratory, only about 10 m away from its future installation site. The following spring, two installers from the manufacturer arrived for the installation. The construction consisted of a 500 t press on top of which the pressure chamber, a four-shell vessel with 41 mm inner diameter, was placed. In addition, all the necessary pumps were combined on one platform. There was a low-pressure high performance pump for the hydraulic oil for the test preparation and post processing. This was used,

for example, to push the lower piston with its seal set to its position immediately below the lateral gas inlet. Then, there were two plunger high-pressure pumps to move the lower piston into the pressure chamber after the automatic stopping the gas pressure pumps to reach a maximum pressure of 2.5 GPa. The third set of pumps was a two-stage high-pressure diaphragm compressor, which compressed the working gas from 10 MPa to 350 MPa and filled it into the chamber via the lateral gas supply. The third assembly was a large control cabinet containing all nova Swiss valves, electrical switches and display instruments for operating the press (Figure 10). For safety reasons, the entire high-pressure equipment, with the exception of this control cabinet, was enclosed in angle steel protective walls covered with 5 mm deep-drawing sheet steel. If 1.3 GPa was sufficient as the final pressure, high-purity argon could be used. However, the pressure chamber was designed for high-purity helium. "High purity" was necessary because hydrogen in particular would have diffused into the inner wall of the pressure chamber, which would have made it brittle. Extreme cleanliness, i.e., the removal of all residues of the surface coating of the metal sealing rings from the last experiment from the inner surface, was necessary. For this purpose, there was a platinum wire furnace, which was further developed in department V with an ultrasonic measuring set-up and in its last version allowed measurements of the propagation velocity of elastic compressional and shear waves up to 2.5 GPa and 1600 °C (Figure 11). To prevent infiltration of the rock samples by the compressed gas, the samples were welded into a stainless steel casing. However, there were also experiments on electrical conductivity and investigation of the deposition of diamond from the gas phase at extreme pressures. In the experiments for the material interpretation of deep seismic profiles, these gas pressure experiments up to 2.5 GPa and 1600 °C formed the upper end after preliminary tests up to 0.5 GPa and 1.2 GPa/500 °C oil pressure. Research carried out at a later date confirmed the unique commercial possibilities on a world scale for this pressure chamber. During the 13 years of operation of the high performance gas pressure chamber in the ZIPE, there was not a single accident [37–52].

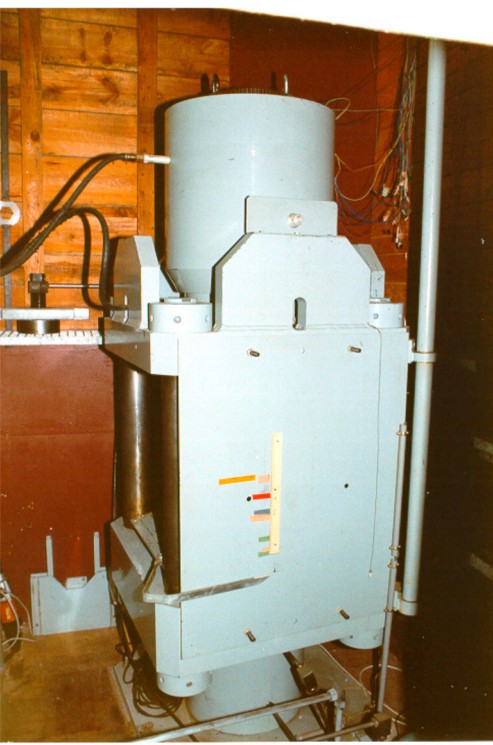

**Figure 9.** 2.5 GPa gas pressure vessel.

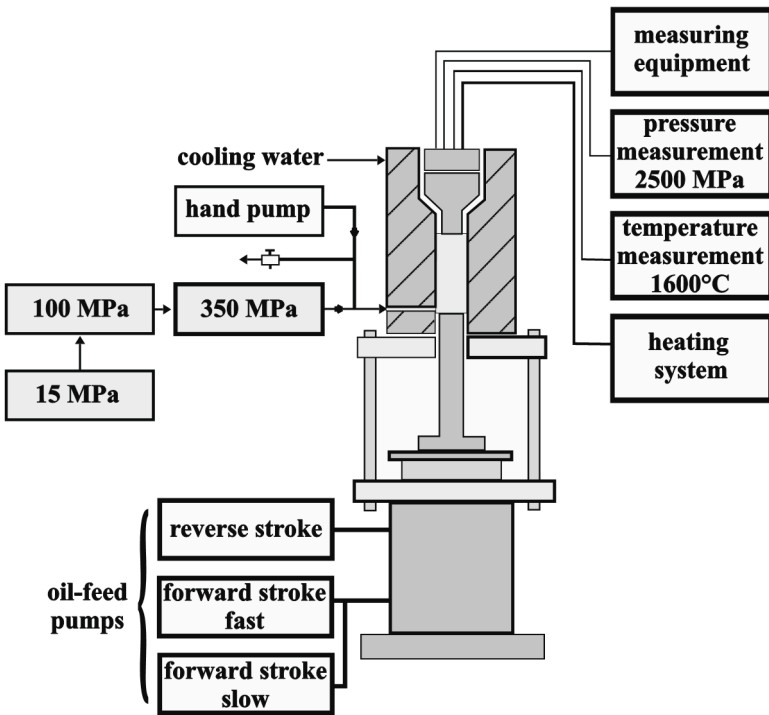

**Figure 10.** Scheme of the 2.5 GPa gas pressure vessel.

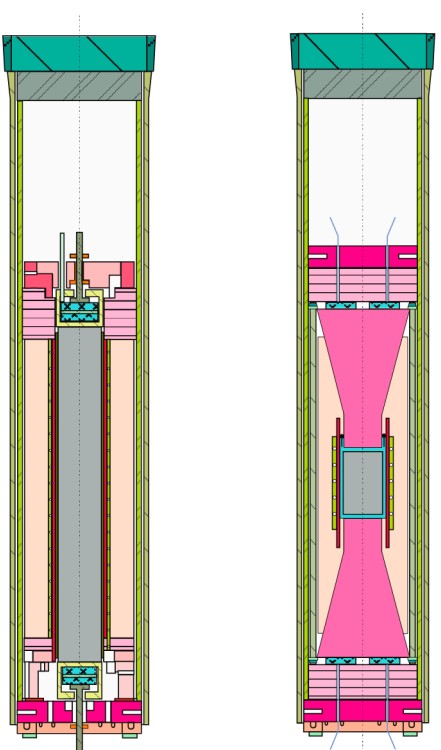

**Figure 11.** Two different measurement devices with sample encapsulation for the 2.5 GPa gas pressure vessel.

In the early 1980s, a new building for department V was also urgently required. This was a two-storey building with no basement and a high-pressure hall with hand washbasins at every room. This accommodated a Czech isostatic press; i.e., an extremely large oil pressure chamber. It could compress about 20 l of oil to 0.4 GPa. For safety reasons, the whole equipment was assembled below

ground level. Therefore, the first part of the new building was the "basement", i.e., a room sized space with massive concrete walls. Only after assembling the press in this pit was the whole building established above. The purpose was to operate the split-sphere high pressure chamber (Figure 12) developed by the company itself following the publication of Kawaii and Endo (1970) [52]. Six steel anvils were combined to provide an outside sphere; the inner part is cube-shaped. It was filled by eight tungsten carbide anvils with a small octahedron sample in the center. After assembling the whole sphere was imbedded in a polysiloxane rubber jacket to prevent oil infiltration. The split-sphere chamber is the moulding of the later multi-anvil chambers. For a first test of the new chamber, the help of the Institute for Marine Research in Warnemünde was called upon. There was a 0.2 GPa plant working with water as pressure transmitting medium. The test was successful, but did not yet reach the target pressure of 1 Mbar. Additionally, in the mid-1980s a 1000 t press and an up-to-date 400 t press (see Figure 13) were installed in the new high-pressure hall. However, the new building with the high pressure hall with the subsurface installation had another advantage. Now the high pressure staff had a party room (see Figures 14 and 15)

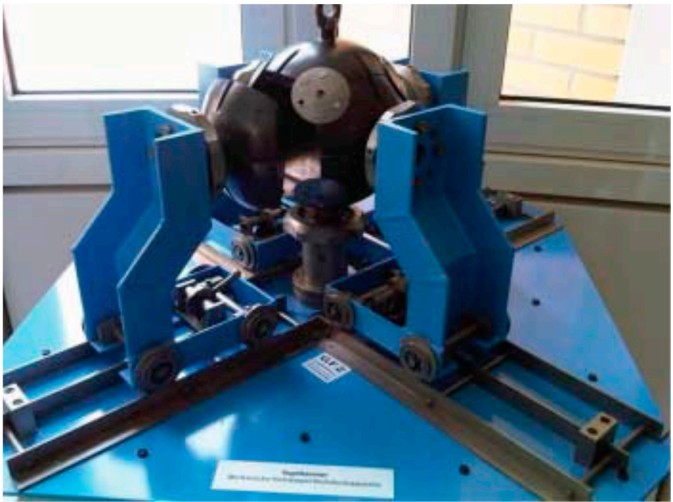

**Figure 12.** Split-sphere high pressure chamber.

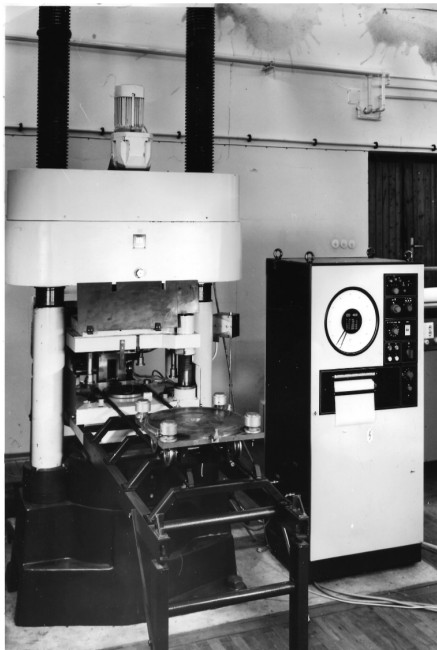

**Figure 13.** New 400 t press.

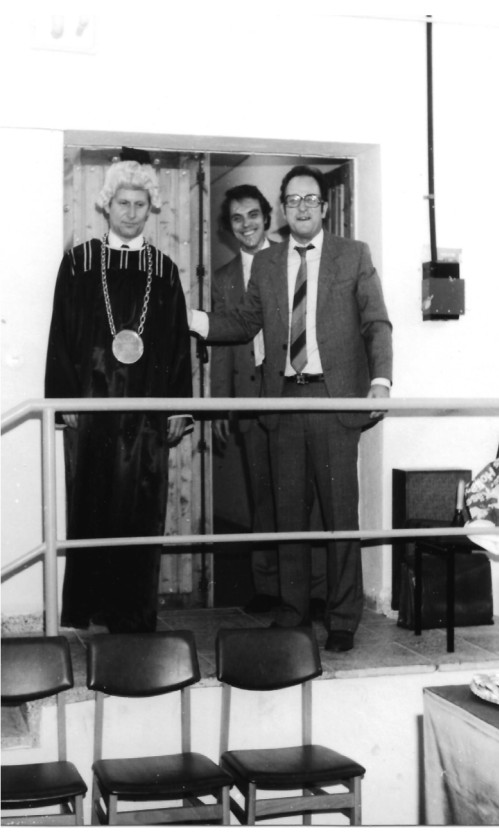

**Figure 14.** The party at Vollstaedt's habilitation (left, central Mueller, right Waesch).

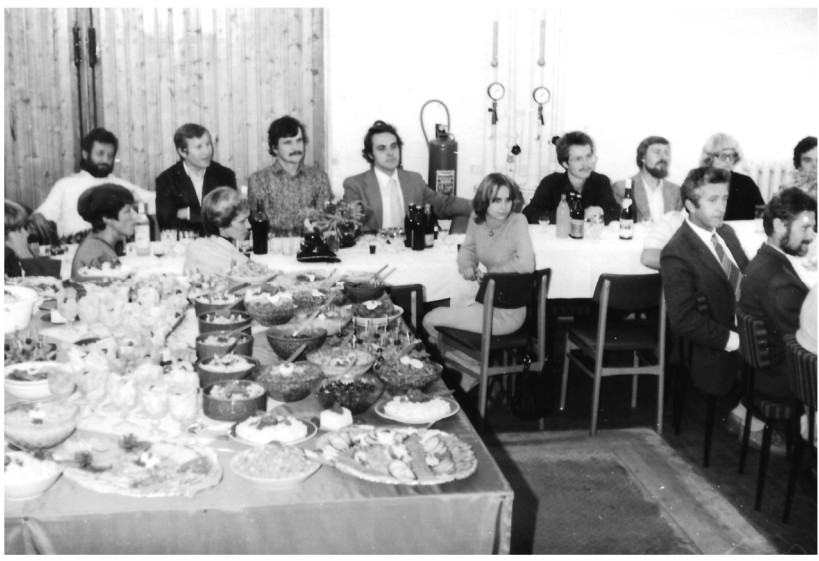

**Figure 15.** The colleagues celebrate this habilitation.

*2.2. Electrical and Structural Relations in the Fe-Ni-S System at High Pressures and Temperatures and the State of the Earth Interior*

The theoretical investigations of the state and structural behaviour of the deeper interior of the Earth required an extended electrical technique, which allowed a reliable extrapolation for an interpretation of the material behaviour to the outer core of the Earth (see Figure 16). The results were published among other things at the meeting of the European High Pressure Research Group at

Potsdam (see Figure 17). The material system investigated consisted mainly of the elements relevant to the Earth's core: iron, nickel and manganese in combination with Sulphur [53–71].

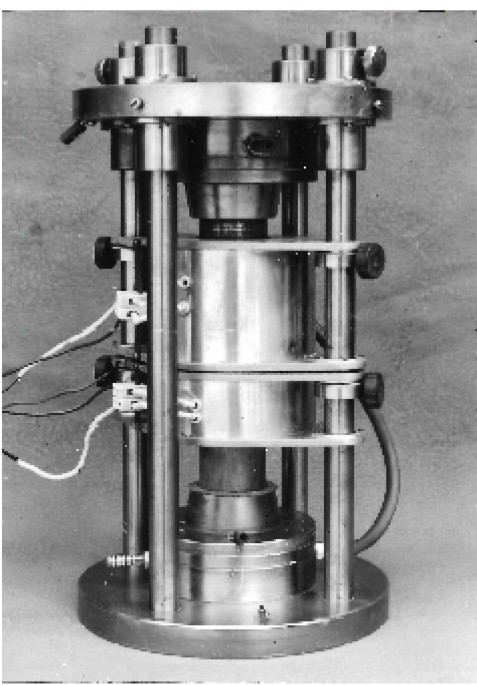

**Figure 16.** Bridgman device for electrical investigations.

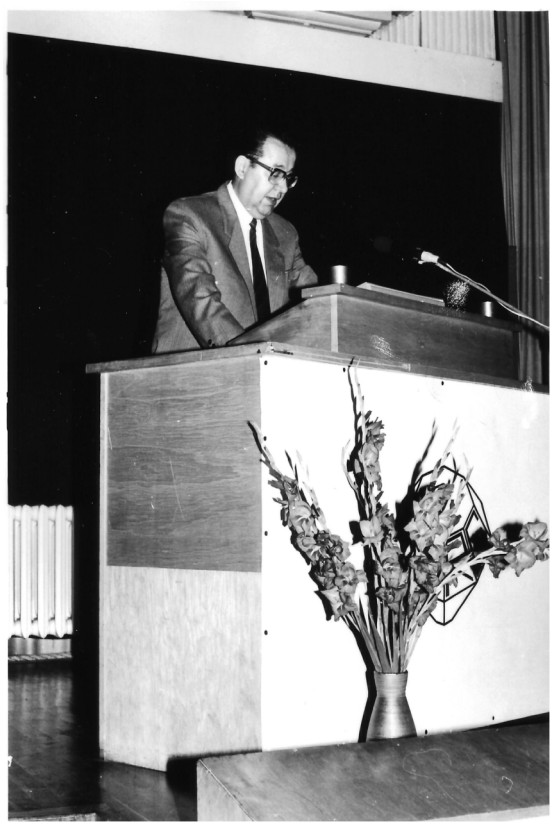

**Figure 17.** Prof. Stiller gives a talk about the Earth core structure at the European High Pressure Research Group 1987 in Potsdam.

The verification of the pressure and temperature stability of the various phases of the elements mentioned or their sulphides and the determination of possible phase transitions under the extreme conditions required the development and manufacture of special high-pressure equipment.

The diamond anvil apparatus (also known as "Squeezer" in ZIPE) and the high-pressure apparatus "Belt" proved to be suitable. Department V especially concentrated on latter, which was later used for industrial diamond synthesis [72–77].

Figure 18 shows a diamond anvil apparatus developed and manufactured at the Central Institute for Physics of the Earth at Potsdam. Phase transitions in the range of more than 10 GPa could be produced in a very small space. Temperatures of more than 1000 °C could be reached by a special laser heating technology. The advantage of this high-pressure technology was that no additional press was required.

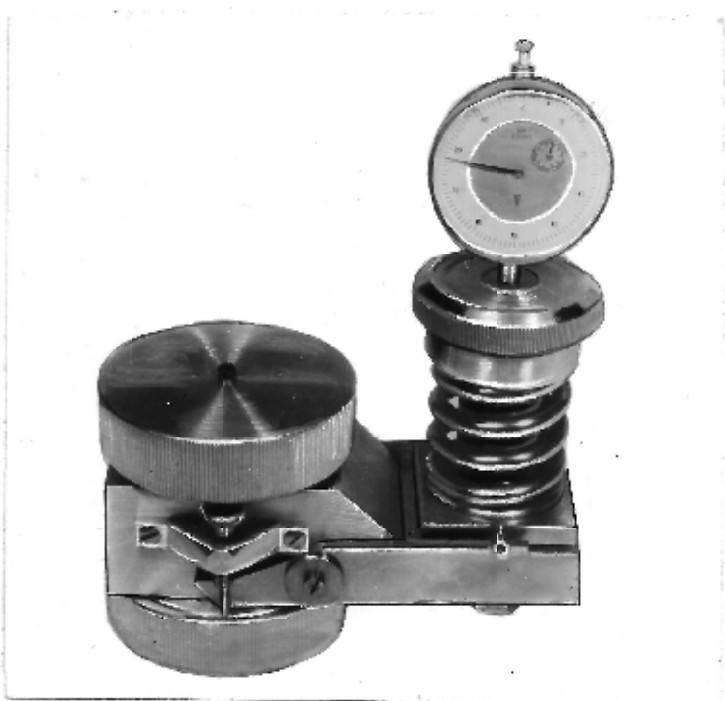

**Figure 18.** The squeezer (diamond anvil apparatus).

For these experiments the Fe-Ni-S-system was selected. X-ray diffraction measurements up to 21 GPa were carried out and phase transitions at 0.4 and 5.0 GPa were detected. In comparison with density calculations, derived from shock wave experiments, the Fe-Ni-S-system could be a candidate to discuss the composition of the inner Earth.

The "Belt" type high-pressure chamber (Figure 19) was a further development of a known variant with regard to a split support ring system and reached pressures of up to 8 GPa at temperatures of up to 1800 °C. For the "Belt" suitable presses according to their size were necessary. The advantage was the relatively large working volume and the fact that the design of this chamber enabled quasihydrostatic conditions, which led to optimum pressure and temperature gradients during the experiments.

### 2.3. Phase Transitions at High Pressure and Temperature–the System Graphite-Diamond

After the successful synthesis of industrial diamonds in the USA and Sweden in 1956, in the Soviet Union in 1961 and in the ČSSR (Czechoslovak Socialist Republic) in 1965 on the basis of various high-pressure technologies, the first tentative proposals to include this economically important technology in the research spectrum were made in the GDR as early as 1966. Such ideas were not supported by leading industrial companies (e.g., VEB Carl Zeiß Jena) or government officials (VEB – nationally owned company). Politicians referred to the agreements within the framework

of the COMECON, according to which the Soviet Union, as a producer of industrial diamonds, was responsible for the entire COMECON sector.

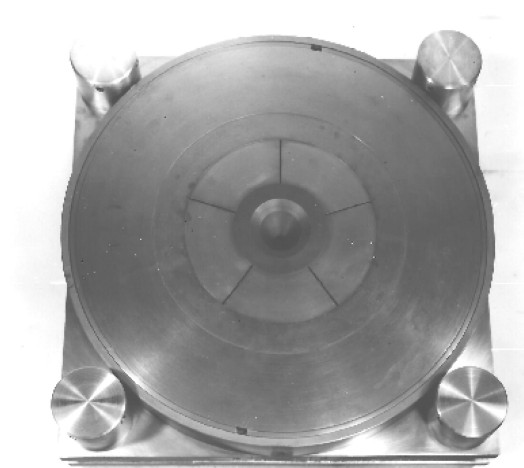

**Figure 19.** Split belt ring system.

At the Central Institute for Physics of the Earth, Potsdam, high-pressure technology had been so developed at 1978 that nothing gets in the way of initial attempts at diamond synthesis. Contrary to clear instructions from the Presidium of the Academy of Sciences of the GDR, successful synthesis experiments to prove the pressure and temperature parameters were carried out at Potsdam in 1979. A few years later these results were also protected by patents. Figure 20 shows the investigation of the first successful synthesis experiment [78–81].

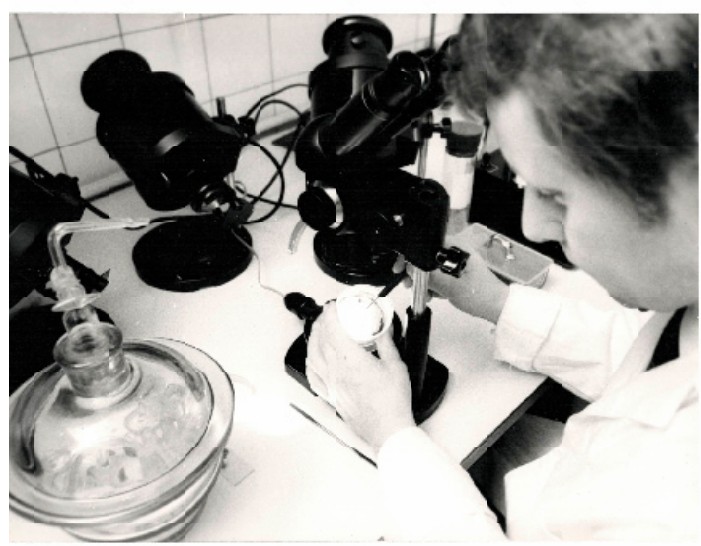

**Figure 20.** Inspection of the first synthetic diamonds by Vollstaedt.

In 1981, the GDR's Academy of Sciences (Zentralinstitut für Physik der Erde, Potsdam, ZIPE, in cooperation with the Zentralinstitut für Festkörperphysik und Werkstoffwissenschaften, Dresden) was surprisingly commissioned to begin research into the production of super-hard materials as part of a state plan topic. This basic research, starting with the synthesis of diamond, was also carried out for the synthesis of the superhard materials cubic bornitride (CBN) and polycrystalline diamond (PKD) and only ended with the transformation of the ZIPE.

Only shortly after the start of the basic research mentioned, companies were founded for the pilot production. Until the year 1989 more than 50,000 carats, diamond powder in different grain sizes were produced here and successfully used in the glass industry as well as in tool manufacturing. These achievements and successes are therefore especially important for as there have been a number of difficulties and mistakes in the accomplishment of the tasks. Materials important in the production process for superhard materials were not available on the GDR market. The usual standard material for the sealing pressure transmission–pyrophyllite-could only be obtained for foreign currencies, as well as the larger high-pressure high-temperature press technology.

After extensive field work to extract pyrophyllite in Vietnam (supported by the Institute for Raw Materials Research, Dresden), a lucky "substitute" was found from the company's own deposits. A deposit near Bockau/Erzgebirge described as pyrophyllite in older works was paragonite. A site survey by the second author revealed the presence of paragonite. After intensive investigations, this material, which was the unused vein material for emery mining at the beginning of the 20th century, proved to be a suitable sealing material for diamond synthesis [82]. Even special processing of the material made it possible to improve the properties of the diamond synthesis process (patent application).

The final decision for the production at the industrial partner Collective Combine Erdöl/Erdgas at Gommern came too late. The import of high-pressure technology and the establishment of diamond production in Gommern did not occur. In addition, the third high pressure hall of department V had a very short life. Here the two northern Korean wound frame presses and a US MTS stiff press were accommodated. During a research visit of the coauthor to the National Institute of Inorganic Materials, Tsukuba, Japan (1989), the cubic phase of aluminum nitride could be detected for the first time (Figure 21).

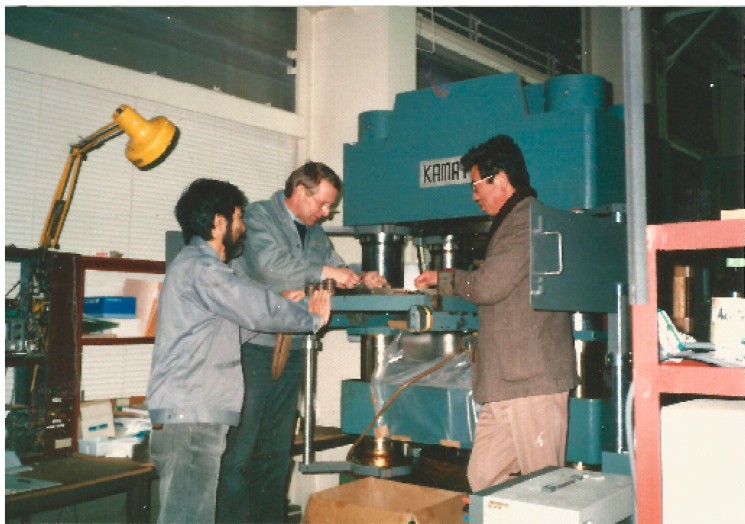

**Figure 21.** Vollstaedt in Tsukuba, Japan (central, left Kanda, right Fukunaga).

*2.4. Planetary Investigations and Special Experiments–Measurements on Lunar Material*

In 1970, 1972 and 1976 the Russian automatic spacecrafts Luna 16, 20 and 24 transported ca. 400 g of material from the lunar surface to the Earth for scientific investigations. The Central Institute for Physics of the Earth in Potsdam received a small part of this lunar regolith (ca. 3 g) for complex mineralogical experiments, together with other Institutes and Universities (see Figures 22 and 23).

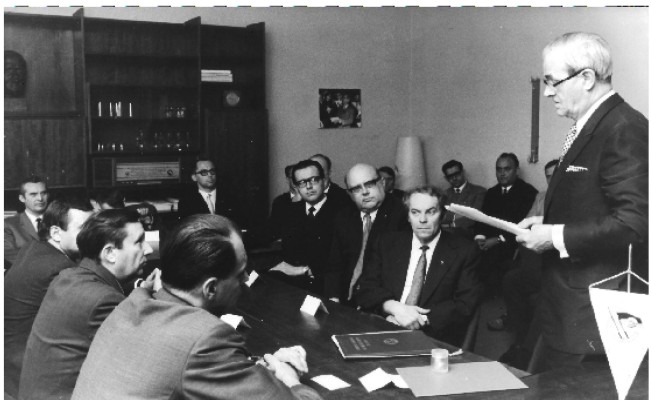

**Figure 22.** Handing over the lunar material (standing Prof. Klare, President of Academy of Sciences, left Prof. Stiller).

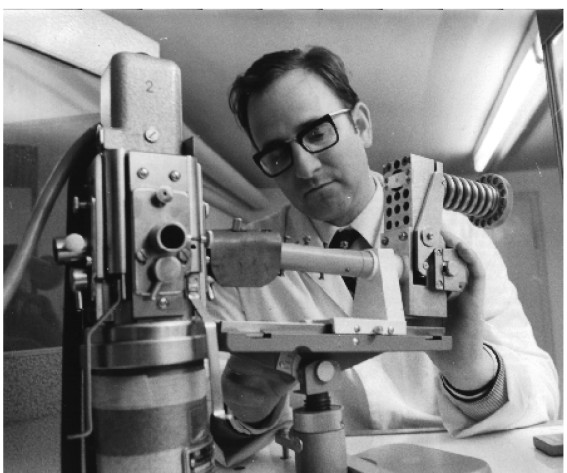

**Figure 23.** Waesch at the squeezer.

The main tasks for these investigations were:

- the determination of the composition and the properties of the lunar regolith and the lunar rocks, helping to explain the lunar history and the planetary evolution, especially the origin of the moon, and
- to develop experimental methods for using very small quantities of lunar materials (mg) to get very exact results.

To determine the composition of the minerals, their structural and chemical behaviour X-ray-spectroscopic investigations have been used. The silicic components of the different rock types were determined.

Petrological investigations on small rock fragments to describe fracture structures together with measurements of the electrical conductivity of the regolith were carried out [83–90].

For the investigation of the very small regolith particles (see Figures 24 and 25) a special high voltage microscope (1 MV) and a transmission electron microscope (up to 100 kV) was used. The most interesting results were:

- The potassium content of the lunar rocks is higher than in the terrestrial rocks.
- The composition of the pyroxene-group of lunar rocks corresponds to the rock composition on Earth.
- The main elements, O, Si, Fe, Al, Mg have similar contents in all lunar samples. Only lunar 16 has higher iron content.

- Lunar 20 has high cobalt content caused by meteorite bombardments.
- Complicated fracture surfaces (Figure 25) on regolith particles could be detected, fragment size (0.02–0.2 mm).

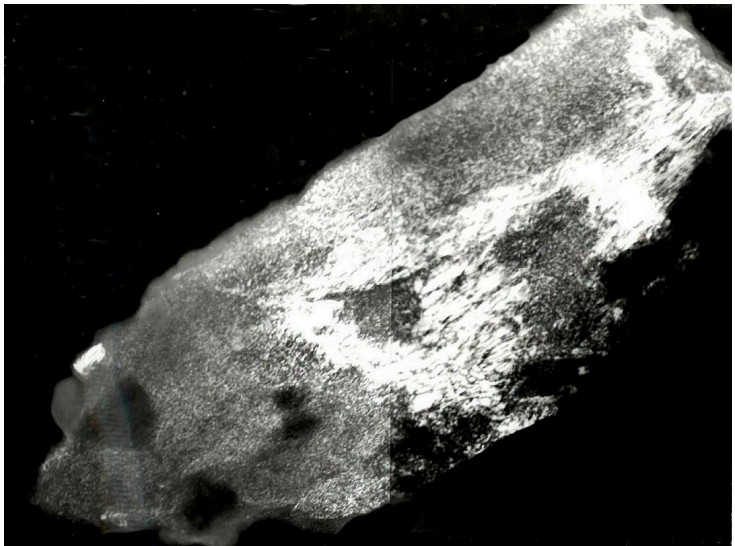

**Figure 24.** Fragment of anorthite (HEM 18.00).

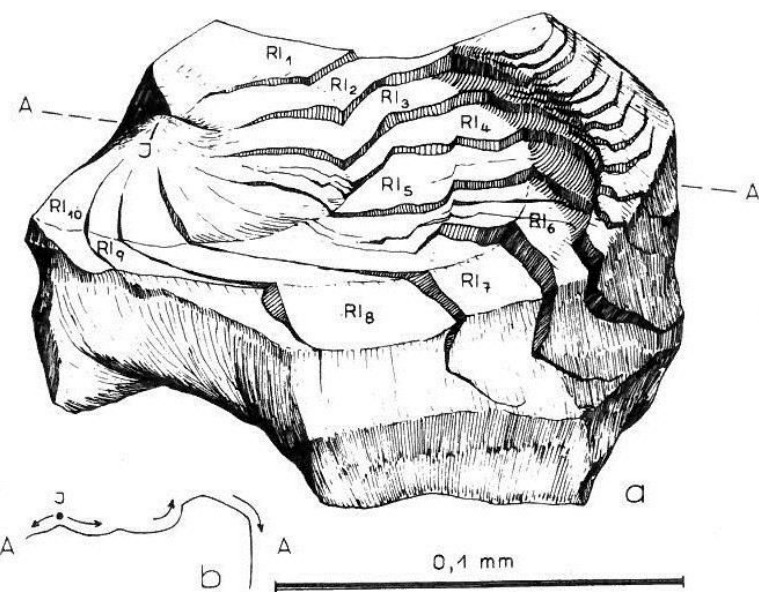

**Figure 25.** Fracture documentation of a small regolith fragment.

## 3. Summary

In a history of about 25 years, the department V of ZIPE with about 40 to 60 employees had made it possible to participate in practically all high-pressure directions relevant for geophysics. The new buildings of the GeoForschungsZentrum Potsdam (GFZ) were established at the place where the three high pressure halls with offices and machine shop had taken over the place. In contact with the demolition or slightly later all high pressure technology (with the only exception of the MTS-press) and all developments were scrapped after the reunification of Germany without necessity. All general high-pressure research was discontinued. At GFZ, three different sections are involved with special aspects of high pressure.

**Author Contributions:** The paper is the result of a joint effort. All authors have read and agreed to the published version of the manuscript.

**Funding:** This research received no external funding.

**Acknowledgments:** The authors highly acknowledge the patience and kindness of all editors just as well especially the support of one of the reviewers.

**Conflicts of Interest:** The authors declare no conflict of interest.

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
