# Peer review of "Complex Geophysical Investigations under Extreme P,T–Conditions at Zentralinstitut für Physik der Erde (ZIPE) (1970–1990)"

_minerals, doi:10.3390/min10050412_

Round 1

Reviewer 1 Report

The manuscript provides an interesting overview about high-pressure research in the former GDR. This review is worth publishing both as a scientific and historiographic document.
Wording and grammar need extensive revision (i give examples below). In particular the introductory section is not comprehensible without knowledge of the German language though it ought to be understandable just in English. Much of syntax, use of tenses, expressions are taken 1-1 from German. Correction is beyond of what a reviewer or editor can provide. The authors should involve somebody with good command of written English in the revision of the text.

The text follows the historic development of this field of research in the GDR. This is an appropriate way of organizing such a review. However, the paper may profit if the authors would provide a bit more context:
In particular for the early stages it would be interesting to know how the research in experimental high-pressure geophysics and material science in the GDR related to similar research in Western Europe, the USA, and Japan - and in the USSR. Were there collaborations? It appears that the initial projects were based solely on domestic resources but in the later years, equipment was acquired from Western Europe, Japan, the USA. Interestingly there is no mentioning of the USSR, which had formidable high-pressure research. Was there no collaboration?

In particular in the early times capabilities were restricted to pressures in 0.1-1 GPa range. Was the research scaled to geophysical problems within this range (lower crust) or rather designed to address deep Earth conditions through extrapolation or studies of analogue systems. The reference suggest that both was done - some explanation within the text would help! What was achieved/published that is worth highlighting in the view of the authors?

Can authors place the research results that were obtained in the ZIPE into context with research in other countries at the same time (Japan, Western Europe, Russia, the USA)? This can be brief. Some of the research cited in the references is follow-up on earlier work (B1-B2 transition in KBr, diamond, CBN synthesis,…). In part this replicate work was driven by economic consideration (diamond, hard material synthesis). In part it is not clear to me why it was (re-)done. Were the earlier studies not accessible? Was ‘originality’ of research not a criterion for publishing? This is not meant as criticism but some explanation by the authors could be useful within the context of this review. How would the authors judge the originality of geophysics research in the GDR?
Finally, explain the abreviation 'GDR'. It may no longer be known to everybody and over time may become even less familiar.

Detailed comments:
Title:
. p,T

P should be capital letter.

. ‘The development of the geophysical high pressure research in the former GDR has been described‘
 Wrong tense! -> IS described here.

.
14 ‘A review of the joined research fields, as experiments on lunar material and Earth 

.  core/mantle material has been described. ‘

same comment!

20 ‘High pressures and high temperatures are as physical parameters usually typical disciplines of physics‘

parameters are not disciplines!

.
‘But in this case the volumes are bigger; pressures over 1GPa are more or less rare. ‘
-> change to ‘… are less commonly applied’

22 ‘in the GDR is that the majority of high-pressure  research was conducted in the field of geosciences. ‘
-> change to proper syntax!

26 ‘ From the mid 1960s to the early 1970s  developments and constellations [1-4]. ‘

In English ‘constellations’ means ‘Sternbilder’!

28 ‘The Geomagnetic Institute (Director: Prof. G. Fanselau) under the direction of F. 

 Frölich have been established a rock magnetic department. ‘

AND
36: (Director: Prof. W. Sponheuer) under the direction of W. Uhlman 

-> If you mention the administrative hierarchy you should explain it - if Sponheuer and Fanselau were directors, why and how was the program directed by Uhlman and Froelich? I guess they had different responsibilities?
35 ‘ …strong theoretical group have been established itself in the Institute for Geodynamics ‘

-> change to ‘had been’ or ‘was’, remove ‘itself’

37: ‘…the deeper interior of the earth and the material behaviour under extreme 
 pressure and temperature conditions ‘
Clarify that you talk about state of the art high P techniques of that time. Compared to the pressures in the transition zone, lower mantle, or core, the pressure ranges of your devices were rather low. Also, in 1981 Mao and coworkers already reached 100GPa level pressures and Akimoto and coworkers worked with large volume presses up to 10 (20?) GPa. Perhaps it would help to clarify the scope of the work in thelab’s you describe here: Lower curst? Upper mantle?

. 78  Additionally later a 1,000 t press was installed in the new high-pressure hall and a modern 400 t press 

change to ‘later a 1,000 t press and a modern 400 t press wERE ALSO installed in the new high-pressure hall…’
What does ‘modern’ mean? ‘Later ‘ is when? Simply give the years of installation.

‘the material behaviour in deeper layers of the Earth (see Figure 12). ‘
-> Figure 12 shows a photo of Prof. Stiller
Also - what does that mean: ‘deeper layers’ - you could not reach core conditions. Did you plan to constrain core conditions through extrapolations or did you focus on lunar or planetesimal cores (or Mars)?
You mention the Earth core further below - it may be useful to explain the scopes and give a brief overview of the results.

Figure 12. caption: ‘ H. Stiller gives his talk at the EHPRG at Potsdam. ‘
‘… gives a talk…’

‘…pressure and temperature stability of the elements…’
-> the elements are stable - you mean their various phases!

. At the Central Institute for Physics of the Earth, Potsdam, high-pressure technology had been so 

. 240  developed since 1978 that nothing gets in the way of initial attempts at diamond synthesis. Despite 

. 241  clear instructions from the Presidium of the Academy of Sciences of the GDR, successful synthesis 

. 242  experiments to prove the pressure and temperature parameters were carried out at Potsdam in 1979. 

-> This entire phrase is quite confounded - please clarify!

253: ‘…mentioned firms were found for the pilot production. ‘

-> companies not ‘firms’!

257: ‘Important material conditions for a successful synthesis were not available on the GDR market. ‘

-> this is an interesting statement but the wording is not clear!
What does that mean exactly: ‘important material conditions’?
Rephrase: Materials important in the production process… were not available. For instance, pyrophyllite…

269: ‘Neither  the import of high-pressure technology nor the establishment of diamond production in Gommern did not happen. ‘

Wording! In English there is no double negation for rethoric emphasis. Happen -> occur.

‘The silicatic components of the different rock types have been determined. ‘

silicic/silicate - also :Wrong tense! (should be: were determined)

Author Response

Dear Reviewer 1,

Thank you for all your highly valuable remarks and hints. It was willing to follow all of them. I hope that I did not oversea anything. In addition to many supplements I added a complete new paragraph about the situation of scientists in the GDR.

Yours sincerely

Hans J. Mueller 

Reviewer 2 Report

Overall I find this manuscript as a historical construct very educational to members of the mineral physics community who may not know the history of high pressure research, particularly in Germany. The story is a good one, and its inclusion in the special volume for Anderson will be an excellent addition.

In some ways, it is difficult to judge this manuscript on the merits we traditionally do for scientific articles, so I will endeavor to do my best to view it as a story that lays out the historical journey of high pressure research in Germany for the community.

Again, as I said above, the story is a sound one, and perfect for inclusion in the special issue for Orson. There are, however, quite a few grammatical issues that could benefit from a thorough re-read and edit process. Since this article in particular relies heavily on historical narrative and flow, it is important not to take the reader out of the story with potentially confusing phrasing or awkward statements. (I apologize, but I do not have enough time to list all examples and potential fixes as I am already somewhat delayed in completing the review.) There are several examples of this throughout most of the manuscript, starting in the abstract and introduction all the way to the summary.

A couple of specifics:

GDR is not defined in the manuscript and is just spoken of as though the reader knows what it is. While this is likely true in Germany, I'm humbled to admit that us silly Americans are not as likely to pick up on its meaning. My rule of thumb is to always define an acronym on its first use so as to avoid potential confusion.

The Table is trying to succinctly convey the comparisons between the institutes and their capabilities at the time, but it is somewhat clunky and awkward (again, with a few English issues as well). Would this be better served as a narrative paragraph in the text, or can it be edited to add clarity for the reader?

I'm not used to seeing pressures or loads in units of Mp, and Mp is undefined in the text. Is it Megapascals? If so, it should be MPa.

I'll close the review here to prevent any further delay by saying that I know this is an interesting and valuable story to be added to the special issue for Orson, and that the manuscript is modeled after a similar one written by Bob previously focusing on his experiences over 50 some years. The story is good and I think deserves telling, but the present form requires extensive editing for English and story flow. I would also recommend tying in the pictures and figures a little bit better than they are right now just to keep the story tight.

Thank you for the opportunity to be a reviewer on this article, and if any more is required of me, please don't hesitate to ask. I definitely look forward to seeing the inclusion of a more polished finished product in the the special issue. Good luck!

Author Response

Dear Matt (Reviewer 2),

Thank you very much for deciphering your identity. Your review corresponds to high degree the review 1. So by following review 1 I also followed yours. Sorry für the confusion about the units - MPa, Mp etc. Especially the Megapond seems to be very unusual for US. So I had replaced them by the usual tons.

Kind regards,

Hans

Reviewer 3 Report

In the manuscript the authors try to describe the high-pressure research carried out at the Zentralinstitut für Physik der Erde during the period 1970 – 1990. The paper does not contain new experimental or theoretical data, neither data interpretation. The text is mainly focused on technological developments of different kind of presses used in the laboratories, but it lacks in the presentation of relevant experimental results.  The writing is generally not clear (the Introduction, for instance) and should be significantly improved. Finally, it surprises me the fact that the authors appear as coauthors of 92/95 references.

If accepted, I would consider this paper as a review rather than a regular article.

Author Response

Dear Reviewer 3,

Thank you very much for your review. Publicating new results was not our goal. Therefore we have so much references. The value is to open a piece of history to be able to learn from mistakes.I hope I was able to fulfill my own goals. A multifold English revision was performed.

Yours sincereely,

Hans J. Mueller
